# Emergency Healthcare Providers’ Knowledge about and Attitudes toward Advance Directives: A Cross-Sectional Study between Nurses and Emergency Medical Technicians at an Emergency Department

**DOI:** 10.3390/ijerph18031158

**Published:** 2021-01-28

**Authors:** Sun Woo Hong, Shinmi Kim, Yu Jin Yun, Hyun Sook Jung, JaeLan Shim, JinShil Kim

**Affiliations:** 1Department of Emergency Medical Services, Daejeon University, Daejeon 34520, Korea; swhong@dju.kr; 2College of Nursing, Changwon University, Gyeongnam 51140, Korea; skim@changwon.ac.kr; 3Jecheon Fire Station, Jecheon 27143, Korea; yyi1229@korea.kr; 4Emergency Medical Center, Pusan National University Hospital, Busan 49241, Korea; 7942112@naver.com; 5Department of Nursing, College of Medicine, Dongguk University, Seoul 38066, Korea; 6College of Nursing, Gachon University, Incheon 21936, Korea

**Keywords:** emergency nurses, emergency medical technicians, advance directive, knowledge, attitudes

## Abstract

This study aimed to explore and compare knowledge levels about advance directives (ADs) and life-sustaining treatment (LST) plans in end-of-life patients between emergency nurses and emergency medical technicians (EMTs). Using a cross-sectional study design and convenience sampling, 96 nurses and 68 EMTs were recruited from 12 emergency medical centers. A survey on knowledge about and attitudes toward ADs was performed using both online and offline methods between November and December 2019. Emergency healthcare providers were conceptually knowledgeable regarding ADs and LST, although approximately half or fewer had knowledge about ADs (such as the legal process for preparation, family or healthcare providers’ role, and the healthcare proxy). The knowledge levels of nurses and EMTs were moderate. Nurses had significantly greater knowledge relative to EMTs about ADs and LST. Positive attitudes of emergency healthcare providers were also moderately low, with nurses having less positive views than EMTs. Significant differences regarding ADs were found, with younger emergency healthcare providers having fewer career years, no personal end-of-life experiences, and less need for ADs having less knowledge. Emergency healthcare providers’ knowledge about and attitudes toward ADs were moderately low, with EMTs demonstrating a greater knowledge deficit and nurses exhibiting lower positive attitudes. Younger and novice providers had lower knowledge, but younger providers had more positive attitudes, implying that professional education and training should begin early in their careers to enhance their confidence for emergency delivery of advanced care planning.

## 1. Introduction

The consequences of therapeutic advances are two-sided: long-term survival and chronic burdens of care [1,2,3]. People with chronic conditions, which often follow the uncertainty of the course of illnesses, have eventually faced the increased need for supportive and palliative care as approaching near death [4,5]. Such burdens can be reduced with advance care planning (ACP) in which individuals actively engage in the continuum of one’s therapeutic as well as palliative care [6,7,8,9]. An advance directive (AD) can be signed where any adult can state their end-of-life (EoL) values, preferences, and/or healthcare proxy while revising or withdrawing it at any time [9,10,11,12]. In South Korea, since a recently enforced Act on the hospice, palliative care, and decisions about life-sustaining treatments (LSTs) (hereafter, Act for LSTs), attention to ACP and ADs has increased, with more people signing one of the two legal documents [13]. Either “Life-Sustaining Treatment Plans” for terminal patients or those at the end stage of life [14] or “Advance Directive on Life-Sustaining Treatment” for any adult aged 19 years or older [15].

Benefits associated with ACP and/or ADs have been well documented [16,17,18,19], while the utilization remains suboptimal or is delayed until death [20,21,22]. Among the barriers to the limited access, professional perspectives around ACP and/or ADs were reported to influence the patient’s decision on beginning such care or implementing the procedure as desired [6,23,24,25,26]. In particular, knowledge about and attitudes toward ACP and/or ADs were modifiable factors for increasing the likelihood of initiating ACP or AD use [23,27,28], while their knowledge and/or attitudes seemed to vary across the professions and were dependent on their areas of expertise [26,29,30]. In one study that reported knowledge and attitudes among the healthcare professionals, more knowledge about and positive attitudes toward the law regarding the withholding/withdrawing of LSTs were noted among palliative care specialists and geriatricians compared to other specialists, while their knowledge deficit was still substantial [30].

Among healthcare providers, emergency healthcare providers who may encounter the circumstance of an emergency decision to initiate or implement an AD should be equipped with the provision of related services [31,32,33,34]. However, a lack of knowledge of the patient and their EoL wishes by emergency healthcare providers could challenge their active engagement of the patient’s EoL care [32,33,34]. Nonetheless, communication with persons with AD carriers and/or families during the emergency care delivery is critical for effective EoL care. The EoL care that is desired is better accomplished with emergency healthcare providers’ accurate knowledge of and positive attitudes toward the ADs, but that information has rarely been solicited.

Thus, we explored the knowledge about and attitudes toward ADs of emergency healthcare providers, involving both nurses and emergency medical technicians (EMTs) working in the emergency room in South Korea. Specifically, this study aimed to (1) explore the specific knowledge about ADs and LST plans of emergency healthcare providers, (2) compare the level of knowledge about ADs for the general and LST plans for the terminal patients and attitudes toward the ADs between nurses and EMTs, and (3) examine demographic differences in knowledge about ADs and LST plans and attitudes toward ADs of emergency healthcare providers.

## 2. Methods

### 2.1. Design and Participants

A cross-sectional study design was used with a convenience sample of nurses and EMTs surveyed from the 12 regional emergency medical centers. A cross-sectional survey on knowledge about and attitudes toward ADs was performed in which nurses and EMTs who provide emergency care in the current practice were recruited and participated in this study.

### 2.2. Measures

Knowledge was assessed regarding an AD on LSTs (7 items) and LST plans (7 items) [35], which were revised by the authors after the enforcement of the Act for the management of LSTs. Each item was constructed on a dichotomous scale (1 = yes; 0 = no) with minimum and maximum scores of each subscale ranging from 0 to 7 and higher scores indicating more knowledge. The knowledge scale scores were computed using both methods, frequency and percent of correct responses for each item and a sum score of each subscale. In this study, the reliability of the scales was assessed by Kuder Richardson (KR) Formula-20: the KR scores for AD on LSTs (7 items) and LST plans (7 items) were 0.68 and 0.74, respectively. 

#### Attitudes toward ADs

Attitudes were assessed using the Korean version of the 16-item Advance Directive Attitude Survey (K-ADAS) [36]. The extent of an individual’s positive–negative posits for ADs was assessed on a four-point Likert scale of each item (1 = strongly disagree; 4 = strongly agree) in four sections [37]. The possible scores ranged from 16 to 64, with a higher score indicating greater positive attitudes toward ADs. Reliability of the K-ADAS was desirable, with a Cronbach’s alpha of 0.80 [36].

The emergency healthcare providers also provided demographic information, using a standard form including age, gender, marital status, religious affiliation, educational level, and years in profession. Nurses and EMTs also provided information about ADs regarding institutional applicability and personal experiences.

### 2.3. Procedure

After the institutional review board (IRB) approved this study, among 35 regional emergency medical centers, the principal investigator (PI, Hong, SW) contacted the executive persons of 12 emergency medical centers, using her networks to recruit the study participations. A cross-sectional survey on knowledge about and attitudes toward ADs (refer to the Appendix A) of nurses and EMTs was then performed using both online and offline methods between November and December 2019. Subsequently, the questionnaires were returned in sealed envelopes. For the offline method, the PI personally visited the three centers and explained the purpose, contents, and procedure of this study to the potential participants. Those who agreed to participate in the study returned their informed consent statement and survey questionnaires. For the online method, an online link with the informed consent statement was sent to the participating healthcare institutions with responses to the online survey questionnaires indicating voluntary agreement to participate in the study.

### 2.4. Statistical Analysis

Descriptive statistics were computed to describe the sample characteristics, including mean and standard deviation for continuous variables and frequency and percentage for categorical variables. The *t*- or chi-square tests were also performed to compare sample characteristics between nurses and EMTs. The correct answers for each item of the AD knowledge were presented with frequency and percentages; comparisons of correct responses for each item between the two groups of nurses and EMTs were also performed, using chi-square tests. The *t*-test was also performed to compare the levels of knowledge about and attitudes toward ADs between nurses and EMTs. Finally, *t*-tests or one-way analyses of variance were performed to examine demographic differences in the knowledge about and attitudes toward ADs of emergency healthcare providers. To identify differences between groups, the Scheffé test was used as a post-hoc analysis. SPSS Ver. 23.0 [38] was used to analyze data, with the level of significance set at *p* < 0.05.

## 3. Results

Two hundred survey questionnaires were distributed, with a response rate of 95.0% (*n* = 190). The final sample consisted of 164 respondents who completed the survey questionnaires with the exclusion of 26 incomplete responses (nurses, mean age = 29.10 ± 5.09, men 24.0%; EMTs, mean age = 27.19 ± 4.58, men 35.3%) (Table 1). Nurses were older than EMTs (*p* = 0.033). More nurses also reported having a religious affiliation than EMTs (*p* = 001). The majority of both groups completed college or baccalaureate (85.4 vs. 88.2%, *p* = 0.853). Mean years of practice of nurses and EMTs were 5.82 and 3.53 years, respectively, with more nurses being significantly longer in practice (*p* = 0.003).

Approximately one-third of nurses and EMTs had a personal experience with EoL care or ADs of families or relatives (35.4 and 25.0%, respectively, *p* = 0.165); more nurses than EMTs reported having experiences with ADs for their family or relatives (*p* < 0.001). The majority of both groups reported the need of ADs, with more nurses having a perceived need for an AD than EMTs (78.1 vs. 57.4%, *p* = 0.004).

### 3.1. Knowledge about ADs and LST Plans

Overall, nurses were more knowledgeable regarding ADs with the majority of nurses showing significantly higher knowledge in all aspects of AD on LSTs and LST plans than EMTs (Table 2). Specifically, for the seven-item AD on the LST category, nurses had significantly greater knowledge in most items than EMTs, except for two items regarding a definition of an AD document (87.5 vs. 82.4%, *p* = 0.358) and healthcare proxy (17.7 vs. 11.8%, *p* = 0.380). In five items, significantly different proportions between nurses and EMTs were noted, with more nurses reporting correct responses with a range of 54.2–87.5%, compared to those of EMTs with a range of 16.2–58.8%. Knowledge deficits emerged in four items regarding an AD as a legal document in both nurses and EMTs, with EMTs reporting correct responses to these items (11.8–33.8%), while more nurses reported knowledge (17.7–58.3%), such as family (correct response, 57.3 vs. 33.8%, respectively) or healthcare providers’ assistance with AD preparation (correct response, 54.2 vs. 16.2%, respectively), healthcare proxy (17.7 vs. 11.8%), and registration (58.3 vs. 36.8%).

For the 7-item LST plans, nurses had significantly greater knowledge in most items of the LST plans than EMTs (Table 2). In four out of seven items, significantly different proportions between nurses and EMTs were noted, with EMTs reporting poorer knowledge with a range of 22.1–63.2%, while more nurses reported knowledge with a range of 53.1–85.4%. Both groups had particularly poor knowledge about the family role for an LST plan’s preparation, with a minor group of nurses (2.1%) and EMTs (5.9%) having a similarly correct answer (*p* = 0.202).

### 3.2. Differences in Knowledge about and Attitudes toward Advance Directives between Nurses and EMTs

All participants showed a moderately low level of knowledge about different forms of ADs, “AD on LSTs” (a mean score = 3.82, range = 0.0–7.0) and “LST plans” (a mean score = 4.02, range = 0.0–7.0) (Table 3). Attitudes toward ADs were considerably low among emergency healthcare providers (mean ± SD = 31.04 ± 5.43). In comparisons between nurses and EMTs, nurses’ knowledge about both ADs was significantly higher than that of EMTs, with scores of 4.42 vs. 2.99 (*p* < 0.001) for AD on LSTs and 4.51 vs. 3.34 (*p* < 0.001) for LST plans. On the other hand, emergency healthcare providers had moderately low attitudes toward the ADs, with EMTs having more positive attitudes than nurses (32.13 vs. 30.26, *p* = 0.029).

### 3.3. Demographic Differences in Knowledge about and Attitudes toward ADs of Emergency Healthcare Providers

Emergency healthcare providers who were older, with a longer career, having personal end-of-life experiences, and more need for ADs were more knowledgeable for both forms of ADs than each counterpart (Table 4). Both nurses and EMTs who were older and had more than seven years of work experience (F = 7.94, *p* < 0.001; F = 12.14, *p* < 0.001, respectively) had significantly higher knowledge about ADs (both an AD on LSTs and LST plan). Both nurses and EMTs who responded as having a greater need for ADs also had greater knowledge of AD and LSTs (χ^2^ = 10.67, *p* < 0.001; χ^2^ = 26.31, *p* < 0.001). Further, in knowledge about LST plans, those who were married and had a religion had greater knowledge of LST plans than each counterpart (χ^2^ = 10.87, *p* < 0.001; χ^2^ = 7.70, *p* = 0.006, respectively). Attitudes toward ADs were significantly higher in those emergency healthcare providers under the age of 25, compared to those who were between 25–29 years and 30 years and older (F = 3.72, *p* = 0.026).

## 4. Discussion

Emergency healthcare providers often confront patients in need of ACP or ADs or the skepticism for implementation of one’s desired care stated on an AD. Our study found knowledge about ADs of emergency healthcare providers was low, particularly in areas of the legitimate process for its preparation, healthcare proxy, and families’ or healthcare providers’ roles. EMTs’ knowledge was lower than that of emergency nurses. Further, both groups also showed moderately low levels of attitudes toward ADs, with nurses’ attitudes being less positive compared to those of EMTs. Emergency healthcare providers who were younger, had less than three years of career experience, no personal experiences for ADs, and more need for ADs had lower knowledge regarding ADs. At the same time, younger providers were less positive about ADs compared to each of their counterparts. Initial insights of emergency healthcare providers into ADs imply that education is needed to increase their knowledge and enhance positive attitudes, particularly targeting novice providers. Their enhanced knowledge and attitudes are more likely to facilitate communication with AD carriers during emergency care delivery.

Healthcare providers’ knowledge and attitudes toward ADs are important modifiable factors [24,26] that could facilitate the patients’ decision about and enhance the effective implementation of ACP and ADs [23,28,39]. Emergency healthcare providers were challenged for the implementation of end-of-life care of terminal patients at the emergency department concerning an AD itself, its availability in medical records [40], accuracy, and stakeholders’ disagreements [41]. However, knowledge about ACP and ADs among emergency professionals remains rarely solicited. Thus, we explored knowledge of nurses and EMTs engaging in emergency care and found poorer knowledge among the EMTs than nurses in both legal forms of ADs, an AD on LSTs, and LST plan in South Korea. Specifically, knowledge about each seven-item “ADs” and “LST plans” criteria was considerably low, with less than half of emergency healthcare providers indicating correct answers to four and two items each. The conceptual definition of both forms of ADs was quite well understood among the majority, while knowledge deficits were considerable regarding family or healthcare providers’ roles in the legal process to prepare an AD, a healthcare proxy, and understanding various types of ADs (i.e., LST plan vs. DNR).

The limited evidence prevents emergency healthcare providers’ knowledge about ACP and ADs to be compared to that of other professionals in previous studies. Different measures of knowledge also prevent comparing our results and previous findings. Further, a wide variety in the level of knowledge about ACP and ADs exists according to the areas of practice or specialty/expertise [26,29,30]. Nonetheless, in a systematic synthesis of three survey studies that examined nurses’ AD-related concepts, approximately 60% of knowledge was correct among nurses, with the knowledge of critical care nurses being higher than that of oncology and emergency care nurses [42]. Consistent with our results, Taiwanese nurses who provided care for patients with chronic illnesses showed a moderately low level of knowledge about ACP, with a correct response rate of 53% [24]. In particular, nurses’ knowledge about legitimate process for preparation of ADs and healthcare proxies was not yet well recognized [24,26,43], while conceptual knowledge and application for ADs among mental health professionals were high [26]. In another study, knowledge about ADs of medical (45%) and nursing professionals was also moderately low [27], while among the medical specialists, legal knowledge about withholding/withdrawing LSTs of palliative care specialists and geriatricians was higher than that of other specialists [30]. In summary, knowledge about ADs of emergency healthcare providers in this study of nurses and general medical professionals was relatively low [24,26,42], compared to palliative care specialists and geriatricians [30]. It is more likely that the use of ACP or AD and its effective administration is facilitated through enhanced emergency healthcare providers’ knowledge, thus supporting the need for professional education and training.

Our study also added information about AD attitudes among Korean emergency healthcare providers, who showed a moderately low level of AD attitudes (mean, 31.04; range, 0–64). In past studies, other healthcare professionals mostly had positive attitudes toward ADs [24,26,27,29,30,44]. For example, mental health professionals who comprised physicians, nurses, auxiliary nursing care technicians, psychologists, and social workers had very positive attitudes regarding AD (mean, 80.53; range, 0–90) [26]. In a comparative study [29,30], nurses and physicians showed very high positive attitudes, while nurses’ attitudes (mean, 19.5; range, 0–27) were more positive than those of physicians (mean, 15.1; range, 0–27) [29]. Interestingly, compared to EMTs, nurses had more knowledge but less positive attitudes. Consistent with our result of low knowledge but a positive view of ADs, in one study, knowledge of medical and nursing healthcare professionals was low (mean, 9.31; range, 0–18), but their attitudes toward the ADs were very positive (mean, 75.37; range, 0–90) [27]. Further, attitudes toward ADs of Taiwanese nurses who cared for chronically ill people were moderately positive, despite suboptimal knowledge about ADs [24]. More research studies are warranted to determine the relationship between knowledge and attitudes among emergency healthcare providers.

Possible reasons of both knowledge deficits and low positive attitudes in this sample could be that compared to ample evidence from Western countries, related reports have just begun in Asian countries. In South Korea, since a recently enforced Act on management of LSTs in 2018, healthcare institutional policy and clinical protocol are in the process of development. Thus, emergency healthcare providers may not yet be in this loop of clinical provision process. Another reason could be associated with their expert areas in that low knowledge of EMTs compared to nurses in this study was possibly associated with limited access of EMTs to an individual’s documented medical history [34], leading to their poor knowledge about legal aspects of ADs. Attitudes of emergency healthcare providers also seemed to be less positive compared to other healthcare providers’ in most areas, including internal medicine. Given that utilization of ACP and/or ADs was more likely with increased knowledge about positive attitudes toward ADs [27,28], it necessitates emergency professionals’ training to increase their knowledge and enhance positive attitudes through their experiences of ACP for relevant people [27,28], through which they may better assist persons with ADs to make a smooth transition to palliative care. Whether the emergency healthcare providers’ training facilitates communication with people with AD carriers during emergency care delivery needs investigation.

### 4.1. Study Strengths and Limitations

Important insights of emergency healthcare providers’ knowledge about and attitudes toward ADs emerged in this study that could be helpful for prompt and effective implementation of ADs. However, this study had some limitations. The convenience sampling method used in this study may have impacted achieving sample representativeness, limiting the generalization of the study results. Thus, verification is warranted using random sampling of emergency healthcare providers in which geographical differences in the knowledge about and attitudes toward ADs are worthy of investigation. A lack of information about emergency healthcare providers’ AD perspective also precludes from making a relative stance, compared to that of other healthcare providers. Thus, it is unknown whether emergency healthcare providers’ AD perspectives are comparable to those of other professions, requiring the use of a comparative study design in a larger sample involving a variety of healthcare professions. Further, with the survey questionnaires, understanding of the emergency healthcare providers’ AD perspectives for emergency AD care is superficial, which could be better understood using mixed methodology.

### 4.2. Implications for Emergency Clinical Practice and Research

This study provides initial insights into emergency healthcare providers’ knowledge about and attitudes toward ADs. Our results imply that professional education and training should target increasing knowledge, particularly of legal procedures and institutional policy, and positive attitudes, particularly to facilitate earlier career development of emergency healthcare providers. This could improve their confidence toward providing care through ACP and ADs and help to ensure the legitimate administration and implementation of an individual’s desired EoL care in an emergency. 

Results of this study also provide directions for future research. More studies are warranted, using cross-organizational data and data including hospital leadership that could be based on developing an effective model for AD implementation during emergency settings and coping with the situation to provide emergency AD care [45,46]. Development of models specific to the emergency AD care needs to be compatible with the legal procedure [47] and organizational policy, thus ground works are also required to prepare the related Act.

## 5. Conclusions

In this initial exploration and comparison of the knowledge about and attitudes toward ADs of emergency healthcare providers, their knowledge was found to be moderately low, with EMTs having a greater knowledge deficit than emergency nurses. Attitudes toward ADs of emergency healthcare providers were also moderately low, with nurses’ attitudes being less positive than those of EMTs. Our results further indicate demographic differences in knowledge and attitudes in that emergency healthcare providers who are younger and have less than three years’ of experience demonstrated lower knowledge. Younger providers, however, had more positive attitudes. 

## Figures and Tables

**Table 1 ijerph-18-01158-t001:** General characteristics of participants (*n* = 164).

Characteristics		Categories	Nurse (*n* = 96)	EMT (*n* = 68)	χ^2^ *(p)*
			*n* (%)	*n* (%)	
Age (years)		<26	22 (22.9)	29 (42.7)	* 6.79 (**0.033**)
		26–29	41 (42.7)	23 (33.8)	
		≥30	33 (34.4)	16 (23.5)	
		Mean ± SD	29.10 ± 5.09	27.19 ± 4.58	
Gender		Male	23 (24.0)	24 (35.3)	2.50 (0.114)
		Female	73 (76.0)	44 (64.7)	
Marital status		Unmarried	71 (74.0)	57 (83.8)	2.26 (0.133)
		Married	25 (26.0)	11 (16.2)	
Religious affiliation		Yes	49 (51.0)	18 (25.4)	10.78 (**0.001**)
		No	47 (49.0)	50 (74.6)	
Educational level		Associate	30 (31.3)	23 (33.8)	0.32 (0.853)
		Bachelor	52 (54.1)	37 (54.4)	
		≥Master	14 (14.6)	8 (11.8)	
Years in profession		<3	34 (35.4)	42 (61.7)	* 11.35 (**0.003**)
		3–6	32 (33.3)	15 (22.1)	
		≥7	30 (31.3)	11 (16.2)	
		Mean ± SD	5.82 ± 5.11	3.53 ± 3.33	
Personal experience	Family or relative death in the past year	No	62 (64.6)	51 (75.0)	2.01 (0.156)
		Yes	34 (35.4)	17 (25.0)	
	AD writing experience of family or relative	No	66 (68.8)	65 (95.6)	17.84 (**<0.001**)
		Yes	30 (31.2)	3 (4.4)	
Experience of EOL care		No	35 (36.5)	31 (45.6)	1.38 (0.240)
		Yes	61 (63.5)	37 (54.5)	
Experience with DNR consent		No	29 (30.2)	11 (16.2)	4.25 (**0.029**)
		Yes	67 (69.8)	57 (83.8)	
Perceived need for an AD		Not necessary	21 (21.9)	29 (42.6)	15.13 (**0.004**)
		Necessary	36 (37.5)	22 (32.4)	
		Very necessary	39 (40.6)	17 (25.0)	

Abbreviations: EMT, emergency medical technician; AD, advance directive; DNR, do not resuscitate; EOL, end of life; SD, standard deviation. Bold indicates statistical significance at *p* < 0.05. *** The *t*-test was used.

**Table 2 ijerph-18-01158-t002:** Comparison of correct answer rate for specific knowledge about an AD on LSTs and LST plans between emergency nurses and emergency medical technicians (*n* = 164).

Categories	Items	Correct Answer, *n* (%)			χ^2^ *(p)*
		Total	Nurse	EMT	
AD	1. An AD can be signed by any adult.	116 (70.7)	76 (79.2)	40 (58.8)	**7.96 (0.005)**
	2. An AD is a document in which an individual indicates desired end-of-life care in advance in case he/she is no longer able to make decisions due to illness or incapacity.	140 (85.4)	84 (87.5)	56 (82.4)	0.84 (0.358)
	3. Family can prepare an AD on behalf of an individual.	78 (47.6)	55 (57.3)	23 (33.8)	**8.79** (**0.003**)
	4. A healthcare proxy as a surrogate decision maker can be designated on an AD.	25 (15.2)	17 (17.7)	8 (11.8)	1.09 (0.380)
	5. An AD can be registered in a designated agency only.	81 (49.4)	56 (58.3)	25 (36.8)	**7.41** (**0.006**)
	6. A physician’s or nurse’s assistance is required to complete an AD.	63 (38.4)	52 (54.2)	11 (16.2)	**24.28** **(<0.001)**
	7. Any changes or revocation is possible when needed.	124 (75.6)	84 (87.5)	40 (58.8)	17.75 **(<0.001)**
LSTPs	1. An LSTP is a document in which a terminal individual indicates desired end-of-life care.	128 (78.0)	79 (82.3)	49 (72.1)	2.43 (0.119)
	2. An LSTP is prepared by an attending physician.	86 (52.4)	59 (61.5)	27 (39.7)	**7.55** **(0.006)**
	3. Terminally ill patients or patients in the dying phase can prepare an LSTP.	129 (78.7)	78 (81.3)	51 (75.0)	0.93 (0.336)
	4. An LSTP cannot be changed once it has been written.	125 (76.2)	82 (85.4)	43 (63.2)	**10.81** **(0.001)**
	5. An LSTP can be written after discussion with the family.	6 (3.7)	2 (2.1)	4 (5.9)	1.63 (0.202)
	6. A DNR (do not resuscitate) order can be used instead of an LSTP.	66 (40.2)	51 (53.1)	15 (22.1)	**15.96** **(<0.001)**
	7. All medical care including analgesics and antibiotics is discontinued on the completion of an LSTP.	120 (73.2)	82 (85.4)	38 (55.9)	**17.69** **(<0.001)**

Abbreviations: AD, advance directive; EMT, emergency medical technician; EOL, end of life; HPC, hospice palliative care; LSTP, life-sustaining treatment plan. Bold indicates statistical significance at *p* < 0.05.

**Table 3 ijerph-18-01158-t003:** Differences in knowledge about and attitudes toward ADs between nurses and emergency medical technicians (*n* = 164).

Variables		Total	Nurse	EMT	*t* *(p)*
		(Mean ± SD)			
Knowledge	AD (7 items)	3.82 ± 1.81	4.42 ± 1.66	2.99 ± 1.68	**29.34** **(<0.001)**
	LSTPs (7 items)	4.02 ± 1.72	4.51 ± 1.45	3.34 ± 1.83	**20.84** **(<0.001)**
Attitudes	(16 items)	31.04 ± 5.43	30.26 ± 5.28	32.13 ± 5.49	**4.84** **(0.029)**

Abbreviations: AD, advance directive; EMT, emergency medical technician; LSTP, life-sustaining treatment plan; SD, standard deviation. Bold indicates statistical significance at *p* < 0.05.

**Table 4 ijerph-18-01158-t004:** Demographic differences in knowledge about and attitudes toward ADs of emergency healthcare providers (*n* = 164).

Characteristics		Categories	*n*	AD on LSTs		LSTPs		Attitudes	
				M ± SD	t/*F(*p*)	M ± SD	t/F(*p*)	M ± SD	t/F(*p*)
* Age (years)		<25 ^a^	51	3.04 ± 1.87	**13.37**	3.14 ± 1.71	**27.16**	2.05 ± 0.28	**3.72**
		25–29 ^b^	64	4.00 ± 1.83	**(<0.001)**	4.16 ± 1.59	**(<0.001)**	1.90 ± 0.36	**(0.026)**
		≥30 ^c^	49	4.40 ± 1.43	**a < b, c**	4.78 ± 1.49	**a < b < c**	1.89 ± 0.35	**a > b, c**
Gender		Male	47	3.89 ± 1.74	0.10	3.96 ± 1.85	0.10	1.87 ± 0.6	−1.57 (0.120)
		Female	117	3.79 ± 1.84	(0.753)	4.05 ± 1.67	(0.753)	1.97 ± 0.33	
Marital status		Unmarried	128	3.69 ± 1.81	3.34	3.80 ± 1.76	**10.87**	1.96 ± 0.33	1.22 (0.228)
		Married	36	4.31 ± 1.72	(0.072)	4.83 ± 1.25	**(<0.001)**	1.88 ± 0.37	
Religious affiliation		Yes	67	4.10 ± 1.77	2.78	4.46 ± 1.49	**7.70**	1.90 ± 0.36	−1.27 (0.216)
		No	97	3.63 ± 1.82	(0.098)	3.72 ± 1.80	**(0.006)**	1.96 ± 0.32	
* Educational level		College ^a^	53	3.98 ± 1.74	2.44 (0.090)	4.09 ± 1.55	2.64 (0.074)	1.92 ± 0.38	0.29 (0.746)
		Bachelor ^b^	89	3.57 ± 1.82		3.81 ± 1.75		1.96 ± 0.31	
		≥Master ^c^	22	4.45 ± 1.79		4.73 ± 1.83		1.95 ± 0.31	
* Years in profession		<3 ^a^	76	3.28 ± 1.79	**7.94**	3.45 ± 1.70	**12.14**	1.97 ± 0.34	0.51 (0.600)
		3–6 ^b^	47	4.06 ± 1.92	**(0.001)**	4.13 ± 1.70	**(<0.001)**	1.92 ± 0.32	
		≥7 ^c^	41	4.56 ± 1.38	**a < c**	4.98 ± 1.29	**a < b < c**	1.91 ± 0.36	
Personal experience	Family or relative death in the past year	No	113	3.56 ± 1.76	**8.206**	3.88 ± 1.76	2.74	1.96 ± 0.34	0.93
		Yes	51	4.41 ± 1.79	**(0.005)**	4.35 ± 1.59	(0.100)	1.90 ± 0.34	(0.355)
	AD writing experience of family or relative	No	131	3.65 ± 1.84	**6.26**	3.87 ± 1.79	**5.40**	1.95 ± 0.34	0.54
		Yes	33	4.52 ± 1.52	**(0.013)**	4.64 ± 1.25	**(0.021)**	1.91 ± 0.35	(0.587)
Experience of EOL care		No	66	3.70 ± 1.95	0.54	3.74 ± 1.90	3.02	1.96 ± 0.37	0.57
		Yes	98	3.91 ± 1.71	(0.465)	4.21 ± 1.56	(0.084)	1.93 ± 0.32	(0.567)
Experience with DNR consent		No	40	3.90 ± 1.95	0.10	4.10 ± 1.60	0.10	1.94 ± 0.32	0.05
		Yes	124	3.80 ± 1.77	(0.758)	4.00 ± 1.76	(0.750)	1.94 ± 0.35	(0.956)
* Perceived need for an AD		Not necessary ^a^	50	2.94 ± 2.15	**10.67** **(<0.001)** **a < b, c**	2.76 ± 2.04	**26.31** **(<0.001)** **a < b, c**	2.01 ± 0.37	1.50 (0.226)
		Necessary ^b^	58	3.98 ± 1.52		4.41 ± 1.23		1.93 ± 0.29	
		Very necessary ^c^	56	4.45 ± 1.43		4.75 ± 1.15		1.90 ± 0.35	

Abbreviations: AD, advance directive; LSTP, life-sustaining treatment plan; EMT, emergency medical technician; SD, standard deviation. Bold indicates statistical significance at *p* < 0.05. ***
^a, b, c^ The one-way analysis of variance (F-test) was used with Scheffé’s method as a post hoc analysis.

## Data Availability

Not applicable.

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
