# Peer review of "Emergency Healthcare Providers’ Knowledge about and Attitudes toward Advance Directives: A Cross-Sectional Study between Nurses and Emergency Medical Technicians at an Emergency Department"

_ijerph, 2021, doi:10.3390/ijerph18031158_

Round 1
Reviewer 1 Report
Thank you for a good read and the information that you provided on a subject that needs more light brought to it as we see today many hospitals being overwhelmed and it is the providers and patients that pay the higher prices for these instances.
Discuss the study as a test and what challenges were found and tweaks may be needed. For example, you may have found that some of the questions in the survey may be more useful with numeric scales in analysis or a qualitative section, or there may be a better way to collect the data from hospitals or include hospital leadership to compare their ideas and perspectives if you were to do a larger scale study.
The discussion section has some conclusions that I would agree with but I do not see a correlation from the data collected on the survey to the conclusions. I would recommend better explain how these conclusions were reached. Also more relation with international references are needed. Please check following references and include it:
- Goniewicz K, Goniewicz M, Burkle FM, Khorram-Manesh A. Cohort Research Analysis of Disaster Experience, Preparedness, and Competency-based Training among Nurses. Plos One. 2021. https://doi.org/10.1371/journal.pone.0244488
- Yoon HY, Choi YK. The Development and Validation of the Perceived Competence Scale for Disaster Mental Health Workforce. Psychiatry investigation. 2019;16.11:816. https://dx.doi.org/10.30773%2Fpi.2019.0140
- Lee JI, et al. Evaluation of an international disaster relief team after participation in an ASEAN regional forum disaster relief exercise.Disaster medicine and public health preparedness. 2016;10.5: 734-738.
- Albarqouni L, Hoffmann T, Straus S, Olsen NR, Young T, Ilic D, et al. Core competencies in evidence-based practice for health professionals: consensus statement based on a systematic review and Delphi survey. JAMA Network Open. 2018;1.2:e180281-e180281.
- Wetta-Hall R, Jost JC, Jost G, Praheswari Y, Berg-Copas GM. Preparing for burn disasters: evaluation of a continuing education training course for pre-hospital and hospital professionals in Kansas. J Burn Care Res. 2007;28(1):97-104.
- Pedersen MJ, Gjerland A, Rund BR, Ekeberg Ø, Skogstad L. Emergency Preparedness and Role Clarity among Rescue Workers during the Terror Attacks in Norway July 22, 2011. PLoS One. 2016;11(6):e0156536.
- Sultan MAS, et al. Nurses’ Readiness for Emergencies and Public Health Challenges—The Case of Saudi Arabia. Sustainability. 2020;12.19:7874. doi: 10.3390/su12197874
Suggests that increasing knowledge is needed. While this probably a true statement it is not apparent how the data from the survey led to this conclusion.
The study highlights the need to provide particularly of legal procedures and institutional policy. You may want to better explain how you came to the conclusion or strengthen this in the literature review.
There is a discussion about the differences in the answers from Nurses and EMT. I think this would benefit from more discussions on the background of these specialties to give insight into their perspectives. For example, I can see the EMT that has to deal with more patients than providers and resources in the field more often then the nurses in the hospital (Especially non ER nurses) may be more skeptical of an organizations overflow capabilities. In a larger study I would recommend that you include hospital leadership in your survey, they tend to have a much more positive (often unrealistic) view of the capabilities which lead to results like those of Memorial Hospital in Hurricane Katrina.
This is an area that needs more research so that we can work more efficiently in times of disaster and medical surges. As an active physician, I would certainly like to see more projects like this take shape that would bring the right attention to systems being better prepared to deal with these low frequency high severity events. Thank you for your work and thank you for a good read.
Author Response
January 19, 2021
Prof. Dr. Paul B. Tchounwou
Editor-in-Chief
International Journal of Environmental Research and Public Health
Dear Editor-in-Chief and Reviewers:
On behalf of my co-authors, I would like to thank you for your comments and recommendations. We revised the paper for grammar and clarity, and we obtained additional editing support from a scientific editing company. We also made specific changes to the manuscript to address your concerns, which are highlighted in red font.
We would thus like to re-submit the attached manuscript entitled “Emergency Healthcare Providers’ Knowledge About and Attitudes Toward Advance Directives: A Comparative Study between Nurses and Emergency Medical Technicians at an Emergency Department” for publication in International Journal of Environmental Research and Public Health as an original research article. The manuscript ID is ijerph-1075178
Again, we thank you for your thoughtful suggestions and insights, which have enriched the manuscript and produced a more balanced and better account of the research. We hope that the revised manuscript is now suitable for publication in your journal.
I look forward to your reply.
Sincerely,
JinShil Kim, PhD, RN
Telephone (82) 32-820-4229, fax (82) 032-820-4201, email: kimj503@gachon.ac.kr
JaeLan Shim, PhD, RN
Telephone (82) 54-703-7804, fax (82) 54-770-2616, email: jrshim@dongguk.ac.kr

Reviewer 2 Report
The authors (AA) aim to explore the specific knowledge about ADs and LST plans of emergency healthcare providers, compare the level of knowledge about ADs for the general and LST plans for the terminal patients and attitudes toward the ADs between nurses and EMTs, and examine demographic differences in knowledge about ADs and LST plans and attitudes toward ADs of emergency healthcare providers. This is an article useful to increase our knowledge of this topic. Addressing the issues below reported could make this manuscript eligible for the publication.
Specific comments:
Title:
Line 3: I suggest to change comparative study with cross-sectional study.
Abstract:
The abstract is quite informative about what the study found, AA should better report how they did it. AA should explain the structure of the survey (main sections).
Line 16: I suggest to change descriptive comparative with cross-sectional.
Methods:
The study methods should be improved. Add the questionnaire as supplementary material.
Line 80: I suggest to change descriptive comparative with cross-sectional.
Lines 108-110: specify the procedure for the off-line method.
Lines 114-115: specify for what kind of variables AA use t- and chi-squared test.
Lines 116- 117: for bivariate comparison did AA use descriptive statistics? How?
Results:
The overall results section should be improved; it is not clear in some points.
Line 123: AA should add the total number of emergency healthcare providers (nurses and EMTs) that work in the 12 emergency medical centres in order to understand the percentage of involved providers and of the responders out of the total.
Moreover, AA could report the geographical distribution of the 12 emergency medical centres. They could analyse the difference among the 12 emergency medical centres, also according to the geographical distribution.
Line 127: there is a typing error (were had).
Table 1: column t or χ2 is not clear. Clarify the table in order to facilitate the reader for understanding the used test and for what data.
Table 4: column t/F is not clear. Clarify the table in order to facilitate the reader for understanding the used test and for what data. Moreover, AA report a<b,c, a<b<c etc, how did AA deduce this? Did AA perform a post-hoc analysis? If yes, specify it, also in the methods section.
Lines 191-194: Did AA compare providers < 25 years with the other? How did AA deduce that these categories were significantly different? Did AA perform a specific statistical analysis? If yes, specify it, also in the methods section.
Discussion:
Lines 225-226: AA could eliminate results already reported in the results section.
Lines 288-290: AA should better explain this limitation.
Conclusions:
The authors should summarise the conclusions taking into account their initial aims and the limitations of their study design. In addition, future studies could be done to strengthen their results.
Author Response

(The authors gave the same response as above.)

Round 2
Reviewer 1 Report
Thanks for the improved version. However I found authors replied better to my comments in the letter instead of the manuscript.
It looks better now but some work could still be done here.
I also think the literature could be still improved, especially with the suggested references that I sent.
Anyway I will let AE to decide about whether or not the current paper and its findings are well described
Author Response
Thanks to your valuable review, our manuscript has improved a lot.

Reviewer 2 Report
The authors have carefully addressed the reviewers' comments. Overall the changes made have improved the manuscript.
Accordingly, I would suggest acceptance after a minor revision.
AA should cite in the methods section the supplementary material, that gives sufficient information about the items of the questionnaire without the Korean version.
Table 4: delete Scheffé and asterisk in the columns title, the indication trough the note and the asterisks on the interested variables is sufficiently clear.
Lines 320-329: I think that the last part of the discussion should be moved to the conclusions.
Lines 331-343: AA could synthesize this part.
Author Response

(The authors gave the same response as above.)
